# Novel Metamaterials-Based Hypersensitized Liquid Sensor Integrating Omega-Shaped Resonator with Microstrip Transmission Line

**DOI:** 10.3390/s20030943

**Published:** 2020-02-10

**Authors:** Yadgar I. Abdulkarim, Lianwen Deng, Muharrem Karaaslan, Olcay Altıntaş, Halgurd N. Awl, Fahmi F. Muhammadsharif, Congwei Liao, Emin Unal, Heng Luo

**Affiliations:** 1School of Physics and Electronics, Central South University, Changsha, Hunan 410083, China; yadgar.kharkov@gmail.com (Y.I.A.); denglw@csu.edu.cn (L.D.); liaocw@csu.edu.cn (C.L.); 2Physics Department, College of Science, University of Sulaimani, Sulaimani 46001, Iraq; 3Department of Electrical and Electronics, Iskenderun Technical University, 31100 Hatay, Turkey; muharrem.karaaslan@iste.edu.tr (M.K.); olcayaltintas@gmail.com (O.A.); emin.unal@iste.edu.tr (E.U.); 4Department of Communication Engineering, Sulaimani Polytechnic University, Sulaimani 46001, Iraq; halgurd.awl@spu.edu.iq; 5Department of Physics, Faculty of Science and Health, Koya University, Koya 44023, Iraq; Fahmi982@gmail.com

**Keywords:** metamaterials, chemical liquids, transformer oil, omega resonator, sensor

## Abstract

In this paper, a new metamaterials-based hypersensitized liquid sensor integrating omega-shaped resonator with microstrip transmission line is proposed. Microwave transmission responses to industrial energy-based liquids are investigated intensively from both numerical and experimental point of view. Simulation results concerning three-dimensional electromagnetic fields have shown that the transmission coefficient of the resonator could be monitored by the magnetic coupling between the transmission line and omega resonator. This sensor structure has been examined by methanol–water and ethanol–water mixtures. Moreover, the designed sensor is demonstrated to be very sensitive for identifying clean and waste transformer oils. A linear response characteristic of shifting the resonance frequency upon the increment of chemical contents/concentrations or changing the oil condition is observed. In addition to the high agreement of transmission coefficients (S21) between simulations and experiments, obvious resonant-frequency shift of transmission spectrum is recognized for typical pure chemical liquids (i.e., PEG 300, isopropyl alcohol, PEG1500, ammonia, and water), giving rise to identify the type and concentration of the chemical liquids. The novelty of the work is to utilize Q factor and minimum value of S21 as sensing agent in the proposed structure, which are seen to be well compatible at different frequencies ranging from 1–20 GHz. This metamaterial integrated transmission line-based sensor is considered to be promising candidate for precise detection of fluidics and for applications in the field of medicine and chemistry.

## 1. Introduction

Sensors play a vital role in the development of many technological applications such as robotics, smart devices, industrial process control, and automation. With the ever fast growing of interesting applications and advanced technologies, the development of efficient and sustainable sensors is becoming more and more important for their inevitable utilizations in various practical fields. In recent years, thanks to the unique features and strong electric and magnetic coupling, metamaterials (MTMs) are receiving considerable attention and promising for building highly sensitive sensors [1]. MTMs are topological designs which have uncommon specifications such as negative permittivity, permeability and refractive index [2,3]. The unique properties of MTMs are obtained from their periodical structure with coupling effect between their unit cells. The very first report on this material is dated back to 1968 following the published work by Victor Veselago [4]. Generally, MTMs can be proposed from various shapes of periodic resonators having capacitive and inductive parts which can provide strong resonance properties. They also have a sensor layer in the resonator part which is able to recognize the resonant frequency shift even when a small electrical change is occurred in the sensor structure.

Due to the unusual properties, MTMs are widely used for some other applications, such as perfect absorber [5,6,7,8], super-resolution [9,10], antenna [11,12], energy harvesting [13,14], and cloaking [15]. In the sensing applications, subwavelength resonators are usually utilized which provide an effective electromagnetic response [16]. The resonator is a metallic open loop with a dielectric gap. This gap can provide strong resonance due to the subwavelength features. When a small electrical variation is detected in the resonator, the resonance frequency is shifted accordingly. Taking full use of these highly sensitive designs can be utilized for monitoring a small number of samples at microwave and terahertz frequencies [17]. 

Recently, several MTM-based sensors have been proposed to detect liquid properties in microwave frequency regime. Amir Ebrahimi et al. [18] developed a MTM-based sensor to detect liquid properties of water-ethanol combination. Another MTM-based sensor was presented by Muharrem Karaaslan et al. for micro-fluid, strain, and rotation sensing. Also, several studies were published in literature emphasizing the use of metamaterial-based microfluidic sensors for dielectric characterization. For instance, Mehmet Bakır et al. investigated square split ring resonators in X-band frequency range (8.2–12.4 GHz) for the detection of transformer oil and microfluidics both numerically and experimentally [18,19,20,21]. Furthermore, multifunction metamaterial sensors have been used for various applications such as humidity, pressure, thickness, density and temperature sensing in the X-band frequency range [22,23]. A transmission line-based metamaterial with rectangular split ring structure for liquid sensing application was designed by Mehmet Bakir et al. [24]. This structure was successfully used to detect the authentic and inauthentic gasoline samples over X-band. Besides, they also claimed that the key in designing the sensor structure is to provide strong interaction with adequate tenability in electrical properties of the testing samples. With the aim of distinguishing liquids via dielectric constant, Zhang et al. [25] proposed a microwave sensor based on complementary circular spiral resonator (CCSR) which operates at 2.4 GHz. On the other hand, a very high-sensitivity microwave sensor made of a microstrip transmission line loaded with a shunt-connected series LC resonator has been investigated by Amir Ebrahimi and his colleagues for microfluidic complex permittivity measurements. Their proposed structure was fabricated and measured using water/ethanol solutions for the confirmation of the sensing principle [26]. Besides, by utilizing the Q factor concept, a microwave sensor was studied for the detection of glucose concentration at the operating frequency of 2, 5, and 7 GHz [27]. In another study, dielectric liquids were characterized by using differential mode microwave microfluidic sensor based on frequency splitting and a microstrip splitter/combiner configuration and split ring resonators (SRRs) [28]. A microwave biochemical sensor based on a circular substrate integrated waveguide (CSIW) topology was proposed for the determination of aqueous dielectric materials. Resonant perturbation method was performed to investigate theoretical dielectric behavior in comparison with the ideal permittivity. Results had shown that the resonance frequency depends strongly on aqueous solvents, whereby high unloaded, Q factor of about 400 was achieved using the circular substrate-integrated waveguide (CSIW) sensor. This proposed structure can be a promising candidate for application of bio sensing and food processing [29]. The metamaterials based on omega shaped resonator have been investigated by other researchers, however their studies are mainly focused on the metamaterial concept [30,31]. In this work, a novel MTM integrated transmission line-based sensor, which operates over wide frequency range of 1–20 GHz and comprises an omega resonator integrated transmission line, is proposed and investigated. The sensor structure has been verified both numerically and experimentally. It was also used to detect the properties of methanol–water mixture, ethanol–water mixture, and different types of pure chemical liquids. Moreover, the designed structure was utilized to identify clean and waste transformer oils. The proposed structure has a simple geometry, which is composed of omega shaped-resonator through which it is possible to adjust the resonance frequency in the desired band. The main novelty is the use of Q factor and minimum value of transmission coefficient S21 as sensing agent in the proposed structure. The proposed sensor structure can be a promising approach to be applied in energy, chemical, medicine, and other industries such as fuel adulteration sensing application and dielectric characterization of fluidics.

## 2. Theory and Structure Design

The MTM integrated transmission line-based sensor structure is designed by using transmission line theory, as illustrated in Figure 1. Full-wave finite integration technique (FIT) based high-frequency electromagnetic solver CST microwave studio was performed for numerical analysis. The proposed sensor structure is composed of three main layers. As shown in Figure 1a, the top layer is omega split ring resonator (OSRR), and its inner space is assigned as the sensor layer. The second layer is a microstrip transmission line placed on the FR4 (flame retardant 4) type dielectric material which is chosen as the substrate of the sensor structure with a relative permittivity of 4.3, loss tangent of 0.02 and thickness of 1.6 mm. The backside of the structure is covered by a copper metal film which acts as a ground plane. 

The microstrip transmission line is properly optimized to form a magnetic coupling effect with the omega split ring resonator and to disclose resonance frequency changes during the numerical analysis procedure. The material type of the omega shaped resonator, transmission line, and the ground plane are defined to be copper metal with a thickness of 0.035 mm and a conductivity of 5.8 × 10^8^ S/m, being consistence with the experiment setups. In the design, we normalized the impedance of the transmission line to 50 ohms to be matched with the ports which have 50 ohms of impedance in both simulation and practical part of our work. In the MTM designing, the selection of the shape and geometrical parameters play important role in sensing performances. The omega shaped resonator in metamaterial-based sensor is chosen due to its high electromagnetic response to the electrical properties change of the materials being placed in the gap of the omega resonator. Even a small fluctuation of permittivity and loss tangent of the material properties and location in the gap results in a resonance shift to high frequencies. That is due to the surround symmetrical properties of the split ring with respect to the origin of the omega resonator. The required dimensions of the resonator and microstrip line were determined and optimized by using parametrical study and genetic algorithm approach in the electromagnetic simulation software.

As shown in Figure 2a, the optimized geometrical dimensions of the omega resonator are a = 10 mm, b = 11 mm, gap (g) = 0.5 mm, c = 2 mm, d = 2.25 mm, while the boundary conditions for the simulation study are depicted in Figure 2b.The inner-circular space of the omega split ring resonator (OSRR) is used as the sensor element, which is open through the whole structure from the top to the ground plate with a thickness of 1.635 mm. This thickness was defined by the parametric study and it is used to put the sample holder and the liquid sample inside.

In the proposed MTM integrated transmission line-based sensor design, two discrete ports are connected to each side of the transmission line in order to observe the transmitted power in a certain frequency range. When the sample is placed into the sensor layer, the value of transmitted power is varied due to the electrical characteristic of the sample. Boundaries conditions are assigned as open add space in all directions for compatibility with real environment, as can be seen in Figure 2b.

To better understand the working principle of the sensor and electrical mechanism of the proposed MTM integrated transmission line-based sensor, when the liquid is dropped onto the sensor layer enclosed by the omega shaped resonator, the effective permittivity of the resonator gap is increased. Consequently, the charge distribution and capacitance of the gap is significantly changed, thereby modifying the inductance of the omega shaped path. The mutual inductance between the resonator and transmission line is therefore affected leading to the interesting change in the transmission resonance, which can be ultimately interpreted by sensing capability of the whole structure. the capacitive and inductive segments along with their equivalent circuit are illustrated in Figure 3. One can see from the equivalent circuit that the proposed transmission line integrated omega split ring resonator behaves like an RLC circuit model, allowing us to extract the resonance frequency of the OSRR. *R*_t_ and *L*_t_ represent the bulk resistance and the main effective inductance of the transmission line, respectively.

*R*_r_ and *L*_r_ denote the resistive and inductive effect of the omega resonator line. Since the transmission line and omega resonator possess inductance value (*L*_t_ and *L*_r_, respectively), magnetic coupling and mutual inductance occurs between them. The resonance frequency in the equivalent circuit diagram mentioned above is affected by two coupling capacitive effect, which are Cr and Cs. These capacitances can be described by capacitive effect of the resonator gap and the sensor layer, respectively.

The value of capacitance of the sensor layer (Cs) can be differentiated with the samples having different dielectric characteristics and it can be expressed by the equation
Cs = (4 a − g) Cpul(1)
where a and g are the average dimension and split gap distance value of the omega split ring resonator. Cpul is capacitance per unit length which can be calculated as [32]

(2)
Cpul = εrc0Z0

where Z_0_ is the characteristic impedance of the line and 
$c_{0}$
 is the velocity of the light in free space. Hence, the total capacitance (Ct) of the overall structure can be calculated by using
Ct = Co + Cr + Cs(3)
where Co is the capacitance effect of the surrounding space and CS is the capacitance effect of the sample placed inside the sensor layer. C_S_ value can vary for different samples due to differences in the complex permittivity characteristics which can be expressed by

(4)
$$\varepsilon_{\mathrm{sample}}={\;\varepsilon }_{\mathrm{sample}}^{\prime }-{j\varepsilon }_{\mathrm{sample}}^{\prime \prime }$$


The impedance of the omega split ring resonator can be defined as

(5)
$$Z_{r}=\;2R_{R}+{j\omega L}_{r}+\frac{1}{{j\omega C}_{t}}$$

where Zr, RR, j, and ω denote overall impedance, total resistance of the split ring resonator (RR = 2Rr) imaginary part, and angular frequency, respectively. 

The resonance frequency and its shifting of the proposed structure can be calculated by

(6)
$$f_{0}=\;\frac{1}{2\pi \sqrt{L_{r}C_{t}}}$$


It is concluded that the resonance frequency of omega split ring resonator depends on the inductive line, capacitive effect of the gap and of the sensor layer (see Equation (3)). Therefore, the sensitivity of the sensor structure can be tuned by these two parameters. It can be noted that the fundamental operation of the sensor structure is related to the interaction between the sensor layer and the transmission line.

## 3. Simulated Results of the MTM Integrated Transmission Line-Based Sensor

In the simulation and experimental design, we normalized the impedance of the transmission line to 50 ohms in order to match it with the ports which have 50 ohms of impedance. The simulated results for the transmission coefficient of the proposed (MTM) integrated transmission line-based sensor has been analyzed and compared to the measured results at the resonant frequencies without sample, both resonance frequencies are observed at near 1.9 GHz. The numerical and experimental results for transmission coefficient at resonance frequency are in a good agreement with the results presented in Figure 4. 

The required dimensions of the resonator and overall structure are determined by using parametric studies and genetic algorithm approach which is a built-in function inside the software CST program used to provide the optimum result. The parametric studies have been realized to observe the effects of the dimensions of the resonator on the resonance frequency of the structure, as demonstrated in Figure 5. From experiences of previous work, transmission line width (W_T_), gap width of the resonator (g), and resonator width (W) are selected as key factors. The resonant frequency seems to be independent with W_T_ before 5 GHz. However, when the W_T_ decreases to 0.8 mm, a deep peak of −7 dB around 5.70 GHz could be observed, as shown in Figure 5a. The effect of the gap width parameter (g) is investigated from 0.4 mm to 0.7 mm, as shown in Figure 5b, where the resonance frequency was noticed to be increased with the increment of the gap width of the proposed resonator. Specifically, a frequency shift of about 200 MHz could be observed. Also, the resonance frequency at 2.1 GHz did not change except when the resonator width W reaches 0.7 mm, and a very high deep peak at 7.5 GHz is observed. It can be concluded that the resonance frequency of the structure is highly affected by the resonator gap.

In order to understand the operating mechanism of the proposed MTM integrated transmission line-based sensor structure, the surface current and electric field distribution are investigated. In the CST (Computer Simulation Technology) software, the distribution is obtained at the resonance frequency of 2.1 GHz for the case of empty sensor layer. It can be seen from Figure 6a that the surface current distribution small amount is concentrated around the omega shaped resonator with mostly distributed on the transmission line. The flowing current in both the resonator and transmission line are in parallel and anti-parallel directions, while the parallel currents control the electric response and the anti-parallel currents control the magnetic response.

Figure 6b depicts the simulated electric field distribution for the proposed MTM integrated transmission line-based sensor structure. It is noticed that the electric field intensity mostly concentrates in the transmitted side (port 2) of the transmission line comparing with that in the other side at the resonance frequency of 2.1 GHz. Moreover, electric field distribution is highly concentrated on the omega resonator (especially capacitive parts of the omega resonator). Hence, the proposed structure is able to sense any small changes in the electrical characteristics of the sample placed in the sensor layer.

## 4. Measurement of the Electrical Characteristics of Chemical Liquids and Sensor Study

The measurements of dielectric characteristics have been carried out by using vector network analyzer (VNA) (Agilent 85070E) dielectric probe kit which shown in Figure 7. The electrical properties of the chemicals (relative permittivity and loss tangents) have been obtained in the frequency range from 2.5 to 3 GHz. The dielectric probe was calibrated by using air and pure water with pre-known electromagnetic parameters in the same frequency range from 1 to 8 GHz [33]. As such, the electrical properties have been retrieved by using Nicolson Ross Weir (NRW) techniques [34,35]. It is worth to mention that almost similar results were obtained compared to those achieved from the coaxial probe approach. In the numerical software CST Microwave Studio, new materials were defined by importing the data files include dielectric properties for each chemical sample. Finally, the sensing characteristics of the proposed structure are numerically monitored by placing the samples in the sensor layer. In the first stage, the ethanol and methanol with pure water mixture are carefully prepared with different ratios (0, 10, 20, 30, and 40%). The measurements have been performed for each chemical sample of ethanol/methanol mixtures with pure water. Moreover ammonia, isopropyl alcohol, polyethylene glycol (PEG 300), polyethylene glycol 1500 (PEG 1500) and pure water are also used as chemical liquid samples. The measurement calibration has been repeated for each specific measurement. The model establishment has been carried out to sense electrical change of the substances in the sensing layer. In this perspective, the model has been established in order to express any change that is happened in the electrical properties. This means that the transmission line was designed to revolve around the liquid reservoir. In addition, optimization for various length of the transmission line has been realized aiming at increasing transmission magnitude in the related frequency range of 2.5 GHz to 3.0 GHz. 

The test results of the electrical characteristics (dielectric constant and loss factor) of the methanol content have been depicted in Figure 8a–c. The measured relative dielectric constants are about 77.5, 67, 62.5, 58, and 55 for methanol concentration ratio of 0, 10, 20, 30, and 40%, as shown in Figure 8a. On the contrary, the dielectric loss factor are found to be increased with methanol content, while both of them are linearly increased with the rise of frequency as a result of increased ionic/molecular rotations. These are seen to be consistence with the results reported in literature [19,30,36]. Furthermore, the electrical properties of the methanol content have also been validated by carrying out extra measurements. 

The highly dispersive behavior of loss tangent in Figure 8c indicates that the electrical characteristic of the ratio of methanol content in water is linear frequency dependent. At high frequencies, the observed larger dispersion for the high methanol concentration ratio may be ascribed to the predominance of intermolecular interaction between methanol and water molecules at high methanol concentration ratio, which suppresses the dielectric relaxation in this frequency range. Hence, the liquid samples are highly sensitive to the frequency change of the applied electric field, leading to the rise of reactive dielectric component. 

The simulated variation of resonance frequency along with dielectric constant versus methanol content is presented in Figure 9. According to the slope of resonance frequency in Figure 9, the rate of this resonance increment is measured to be about 65 MHz per 10% increment of methanol content. It can be clearly seen from the figure that when the density of the methanol content is increased the dielectric constant is almost linearly decreased. Therefore, the proposed omega based resonator can be a promising tool to be used for methanol content monitoring and for quality control assessment of liquid chemicals.

To further validate the proposed MTM integrated transmission line-based sensor, ethanol–water mixture is also investigated in the same frequency range and boundary conditions. The graphs of dielectric constant and dielectric loss factor, obtained for each 10% ethanol increase in the mixture, are illustrated in Figure 10a–c. The measured relative dielectric constant for ethanol content in water are about 78, 74, 66, 58 and 51 for 0%, 10%, 20%, 30%, and 40%, respectively as shown in Figure 10a. It can be concluded from Figure 10 that the same trends of dielectric constant and dielectric loss changes are seen compared to those of the ethanol–water mixture contents. This indicates the proposed sensor is able to identify the chemical liquids with high robustness.

The results obtained for ethanol–water mixture was noticed to be in good agreement with those previously reported in the frequency range of 3–5 GHz [37]. For both sensors, the resonance frequency is linearly increased with respect to the ethanol content. Manufacturers of dielectric ceramics often use the name ‘quality factor’ for the reciprocal of the tan 
σ
. The term ‘quality factor’ is more commonly associated with microwave resonators. Quality factor, or Q factor, is a measure of the power loss of a microwave system. For the microwave resonator, losses can be of four types: (a) dielectric Q_d_, (b) conduction Q_c_, (c) radiation Q_r_ and (d) external, and could be expressed as

(7)
$$Q_{d}=2\pi \frac{W_{1}}{P_{d}T}=\;\frac{\omega_{0}W_{1}}{P_{d}},{\;Q}_{C}=\;\frac{\omega_{0}W_{1}}{P_{C}},{\;Q}_{r}=\frac{\omega_{0}W_{1}}{P_{r}}$$

where W_1_ is the total stored electric energy in the resonator, 
$\omega_{0}$
 is the angular resonant frequency, P_d_, P_c_, and P_r_ represent the power dissipated in the dielectric, conductor, and radiation respectively. The unloaded quality factor Q_u_ is related to other Q-factors by

(8)
$$\frac{1}{Q_{u}}=\;\frac{1}{Q_{d}}+\frac{1}{Q_{c}}+\frac{1}{Q_{r}}$$

where 1/Q_d_ is dielectric loss, 1/Q_c_ the loss due to conductivity of the metallic plates and 1/Q_r_ is the loss due to radiation. 

Loaded Q-factor is defined by,

(9)
$$\frac{1}{Q_{L}}=\;\frac{1}{Q_{d}}+\frac{1}{Q_{c}}+\frac{1}{Q_{r}}+\frac{1}{Q_{\mathrm{ext}}}$$

where 1/Q_L_ is the total loss of the system and 1/Q_ext_ is the loss due to external coupling.

The loaded Q_L_ is obtained from the measured resonant frequency f and half power (–3 dB) bandwidth 
Δf
 of TE_011_ mode resonance.If conduction, radiation and external losses are negligible, then Q_L_ = 1/tan 
δ
. If all conduction, radiation and external losses are negligible, then the loaded Q-factor depends on dielectric losses in the resonant structure. If the resonant structure contains several (N) dielectrics (one of them is sample under test) then the Q-factor due to dielectric losses is related to the dielectric losses in particular dielectric regions by the following formulae

(10)
$$\frac{1}{Q_{d}}=\sum_{i=1}^{N}P_{\mathrm{ei}}{\tan\delta }_{i}$$


(11)
$$P_{\mathrm{ei}}=\;\frac{{∭}_{\mathrm{vd}}\varepsilon_{i}{\left|E\right|}^{2}\mathrm{dv}}{{∭}_{\mathrm{vt}}\varepsilon \left(v\right){\left|E\right|}^{2}\mathrm{dv}}$$

where P_ei_ is the electric energy filling factor for the *i*_th_ dielectric region and tan 
$\delta_{i}$
 is the dielectric loss tangent for the ith dielectric region. V_d_ is the volume of the DR, V_t_ is the volume of the whole resonant structure, 
$\varepsilon \left(v\right)$
 is the spatially dependent permittivity inside the whole resonant structure and 
$\varepsilon_{i}$
 is the permittivity of the i^th^ dielectric region. 

Figure 11 represents the simulated variation of resonance frequency and dielectric constant with respect to the change in ethanol percent. Similar to that obtained for methanol sensing, the dielectric constant of ethanol showed the linearly characteristics. This result concludes that efficient sensing of ethanol content and its purity can be fulfilled with the help of MTM integrated transmission line-based sensor incorporating omega shape resonator.

Furthermore, the sensing capability of the proposed MTM integrated transmission line-based sensor is investigated through measuring the dielectric permittivity of ammonia, isopropyl alcohol, polyethylene glycol 300 (PEG 300), polyethylene glycol 1500 (PEG 1500), and pure water. The reason for choosing these chemical liquids is that it remains challenge to distinguish from each other due to their colorlessness and less intense odor. The measured values of relative dielectric constant and loss factor for the chemical samples are illustrated in Figure 12a,b, respectively. The dielectric constants are obtained to be about 67, 6, 9, 2, 78 for the ammonia, isopropyl alcohol, polyethylene glycol 300 (PEG 300), polyethylene glycol 1500 (PEG 1500) and water respectively. One can clearly see from the results that the dielectric constant of PEG1500 is close to the dielectric property of air and the dielectric constants of the isopropyl alcohol, PEG 300 and PEG 1500 are close to each other. 

The loss factor for each chemical sample has also been determined and observed, as shown in Figure 12b. Interestingly, it is concluded from the measurement results of the electrical parameters of the chemicals that there is an observable difference between relative permittivity of the liquid chemicals. Hence, determination of the type of chemicals and their ratios/concentration in water can be easily and rapidly achieved by microwave techniques without utilizing any chemical analysis or destructive methods. The unique design of the proposed structure is specifically important to detect/identify the type and concentration of materials in spite of closeness in their dielectric parameters. Therefore, the type and/or concentration of the material can be determined according to the previously defined data on a chip/microprocessor. In addition, the sample range to be determined can be further improved if they have different dielectric constants at the operating frequency of the proposed sensor.

Finally, the electrical characteristics (dielectric constant and dielectric loss factor) for the clean and waste transformer oil has been measured by using Agilent 85070E dielectric probe kit via connecting to a vector network analyzer at the same frequency range of 2.3–3 GHz. The results are shown in Figure 13, where the value of dielectric constant of clean and waste transformer oil are about 2.74 and 2.87, respectively, while the dielectric loss factors for both types of oils started at 2.5 GHz are about 0.2 and 0.17 respectively. When the frequency reaches 3 GHz, the dielectric loss factor for clean oil is 0.28 and for waste oil is 0.23. 

## 5. Fabrication of MTM Integrated Transmission Line-Based Sensor and Measured Results 

The MTM integrated transmission line-based sensor structure was fabricated using the LPKF ProtoMAT E33 prototyping machine. As illustrated in Figure 14b, the transmission line is made of copper and two discrete ports (namely port 1 and port 2). The back of the sample holder is protected by a copper ground plane without affecting the sensing mechanism.

As depicted in Figure 14a, The VNA (Agilent PNA-L series) was calibrated by using a special calibration kit with open circuit, short circuit, and 50 Ohm load connectors in the frequency range of 1–8 GHz and 2.5–3 GHz. Two coaxial test cables were connected to the sensor structure, as shown in Figure 14c. For the measurement process to take place, the samples were directly injected into the sensor layer without the use of a sample holder. This method was similar to the numerical ones and results were obtained without applying any liquid filtering through the substrate walls. After the measurement of the sample, the sensing layer has been carefully cleaned and dried. Since the proposed sensor structure is disposable, there is no need to have any sample holder. To validate the simulated results the experimental results for the all chemical samples has been investigated and compared, the numerical and test results for the methanol, ethanol, and different pure chemical samples are depicted in the Figure 15, Figure 16 and Figure 17 respectively, finally the study of transformer oil condition or transmission coefficients (S21) were then performed for the clean oil, waste transformer oil and air both numerically and experimentally, as results shown in Figure 18.

The data files obtained from the measurement of the dielectric probe kit for different methanol contents in water were imported into the simulation program. Samples have been defined as new materials in the simulation software and they were placed inside the sensor layer of the proposed structure. After that, the methanol–water mixtures were numerically tested by using the sensor structure. For each of the methanol water mixture, the transmission magnitude was almost fixed at the same value. However, as shown in Figure 15a, the minima position of the transmission resonance frequency is found to shift towards higher frequency increasing methanol content with ramp of 10%. Figure 15b shows the measured transmission responses of the sensor for water–methanol samples with various water volume fractions. The experimental results are in good agreement with the simulation ones and indicated that our metamaterials integrated transmission line-based sensor are capable of distinguishing small amounts of methanol–water mixture. 

Furthermore, Figure 16a shows the simulated results obtained for ethanol sensing of the proposed MTM integrated transmission line-based sensor structure with the same condition as obtained for the methanol. To validate the simulated results for the water-ethanol sample with various water volume fractions, measured results are plotted in the Figure 16b. The minima position of transmission resonance frequency is inclined to shift towards the right side with rate of frequency about 80 MHz. 

Transmission coefficient spectra of proposed sensor filled with typical pure chemical liquids (i.e., PEG 300, isopropyl alcohol, PEG1500, ammonia, and water) are illustrated in Figure 17. Apparently, the transmission power for the isopropyl alcohol and PEG 300 presents weaker compared to those of other liquids. Similarly, the minimum resonance frequency points for the isopropyl alcohol and PEG 300 are estimated to be at lower frequency than those of other fluidics. This is due to the reverse relationship between the complex permittivity and resonance frequency. Noteworthy, the proposed omega resonator-based sensor has precisely distinguished these liquids. In this way, as in Figure 17a, the frequency difference between resonance points/minima of isopropyl alcohol and PEG 300 was found to be 130 MHz, while that difference for PEG 1500 and water was observed to be 250 MHz. 

To compare the simulated results again we arranged and measured the value of the transmission response for the different pure chemical liquids at 1–2.5 GHz when the sensor layer filled with PEG 300, isopropyl alcohol, PEG1500, Ammonia and water. The results show that the proposed structure can easily detect any chemical liquids putted in the sensor layer. The measured results are similar with the simulated results excepted isopropyl curve may be due to calibration and measuring setup.

Figure 18 demonstrates the results of simulated and experimental transmission coefficient taken for clean and waste transformer oils by the proposed sensor in the frequency range of 1-8 GHz. Clearly, the designed sensor is able to effectively discriminate the clean, waste transformer oil and air although there as a small difference in the dielectric parameters of the oil samples. Besides, the characteristic frequency shift between clean and waste transformer oil is about 63 MHz. also, a good agreement between the simulation and experimental results is obtained over a broad spectral range from 1 to 8 GHz. The small mismatch between the simulated and measured results can be attributed to the fabricated process, soldering, calibration errors, adaptors, cables and non-perfect testing instruments. The frequency shift was observed at every point of the operating frequency range. Moreover, this discriminative frequency range of the designed sensor structure has been found around 2.1 GHz. 

It also should be noted in Figure 18 that there is some alleviation in the value of transmission power at the frequency points around 2 and 4.5 GHz in both measured and simulated results. In the characteristic frequency range between 1–3 GHz, there is a good agreement between the numerical and experimental studies, which can be viably used to detect the type of oil samples and their pureness regardless the proximity of dielectric parameter value.

The result comparison with other works in term of sensitivity and dielectric constant are shown in Table 1 and Table 2, respectively.

## 6. Conclusions

In this work, a new metamaterial (MTM) integrated transmission line-based sensor omega-shaped resonator with microstrip transmission line is proposed both numerically and experimentally to detect the liquid types and their concentration. The simulations are carried out by using finite integration technique (FIT)-based electromagnetic simulation, which is highly accordant with experimental results. The operating principals are detailed through equivalent circuits, and the physical mechanism of the sensor has been investigated by simulating surface current and electric field distribution. In the chemical liquid sensing study, the electrical properties of the methanol–water mixtures, ethanol–water mixtures, ammonia, isopropyl alcohol, polyethylene glycol 300 (PEG 300), polyethylene glycol 1500 (PEG 1500), and water have been measured. In the chemical liquid sensing study, the proposed structure has well distinguished PEG 300 and isopropyl alcohol despite their similar dielectric characteristics. In order to compare and further verify the robustness of the proposed sensor, the sensing study has been carried out upon all mentioned samples experimentally. The simulated and measured results are found to be in good agreement each with other, whereas an approximately 90 MHz and 80 MHz frequency difference in the characteristic resonance could be noticed for 40% concentration methanol and ethanol respectively. For the clean and waste transformer oil, the frequency difference is monitored 63 MHz at the resonance frequencies around 2 GHz. This MTM integrated transmission line-based sensor can be used for precise detection of fluidics and for applications in the field of medicine and chemistry. 

## Figures and Tables

**Figure 1 sensors-20-00943-f001:**
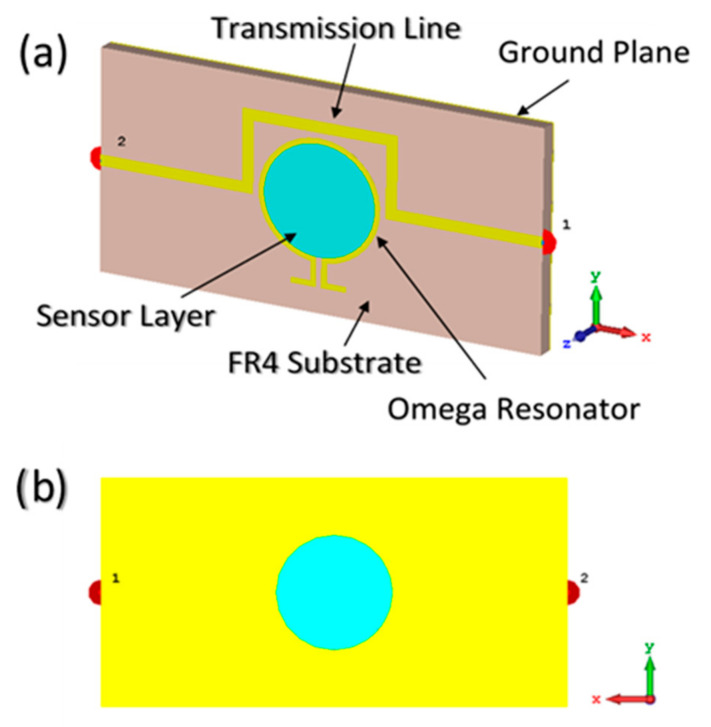
Proposed metamaterial-based liquid sensor design: (**a**) perspective view; (**b**) back view.

**Figure 2 sensors-20-00943-f002:**
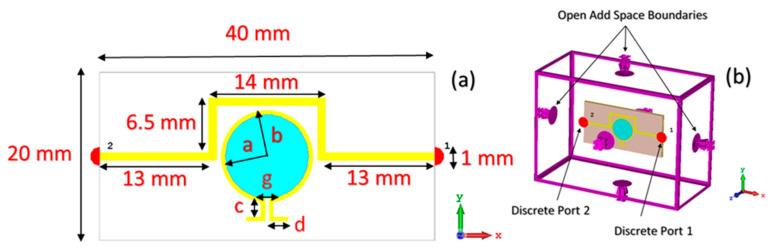
MTM based liquid sensor design: (**a**) design dimensions; (**b**) boundary conditions and added port of each side of the transmission line at simulation program.

**Figure 3 sensors-20-00943-f003:**
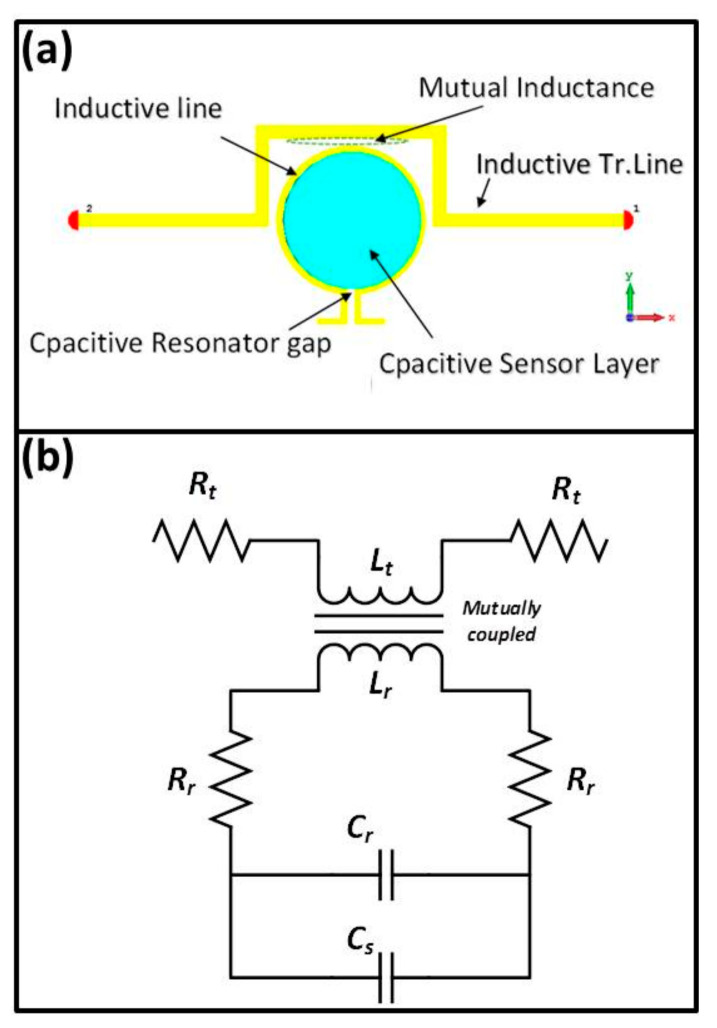
(**a**) Capacitive and inductive part; (**b**) the equivalent circuit diagram of the proposed MTM integrated transmission line-based sensor structure.

**Figure 4 sensors-20-00943-f004:**
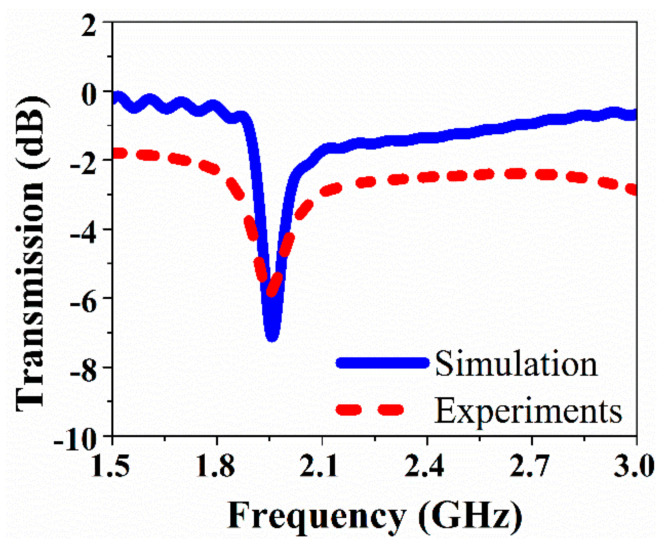
Simulated and measured results of the transmission coefficient when air is present in the sensor hole.

**Figure 5 sensors-20-00943-f005:**
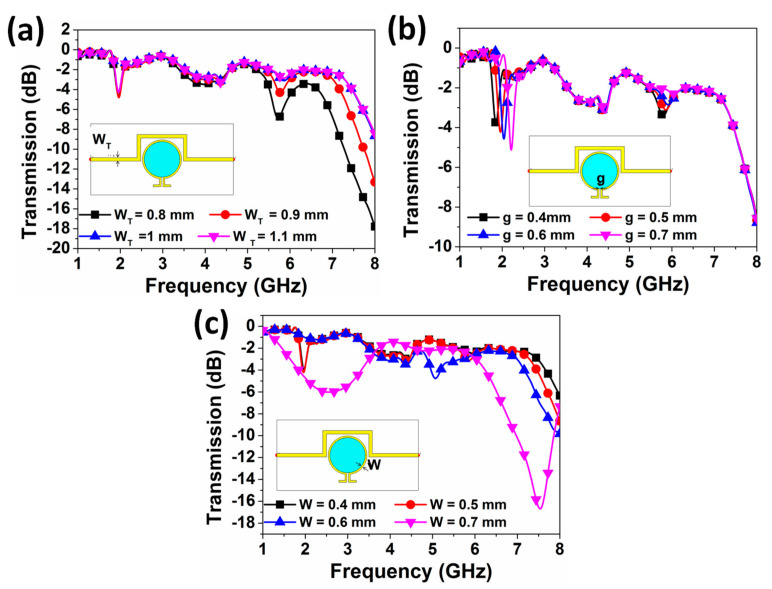
Effect of the variation of (**a**) width of the transmission line W_T_, (**b**) gap of the omega-shaped resonator g, and (**c**) width of the omega-shaped resonator W on the resonance frequency.

**Figure 6 sensors-20-00943-f006:**
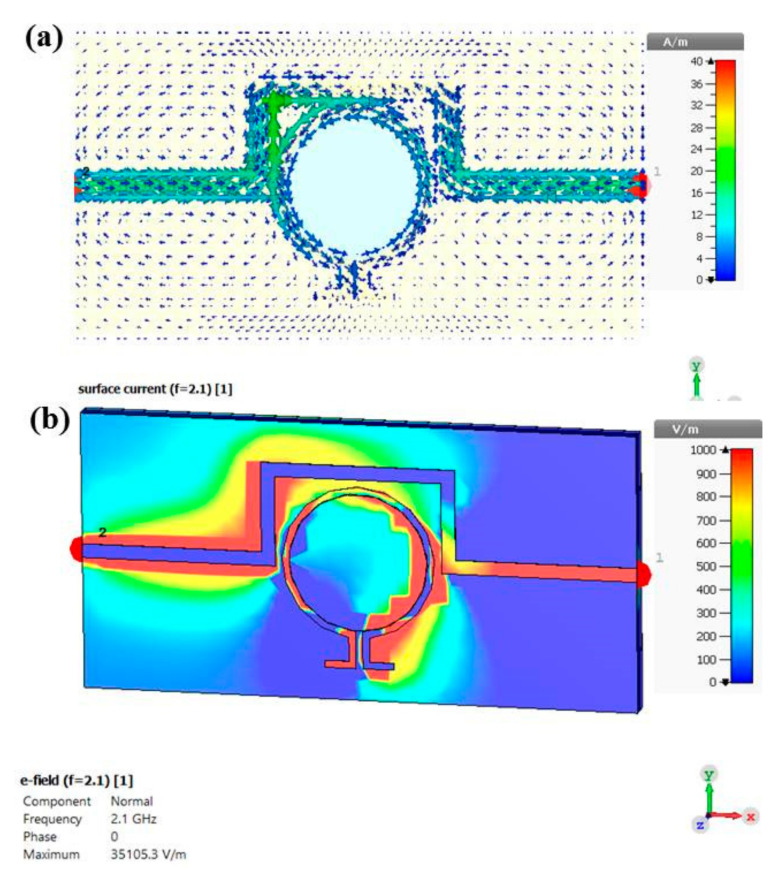
Simulated results for the proposed sensor structure at 2.1 GHz: (**a**) surface current distribution; (**b**) electric field distribution.

**Figure 7 sensors-20-00943-f007:**
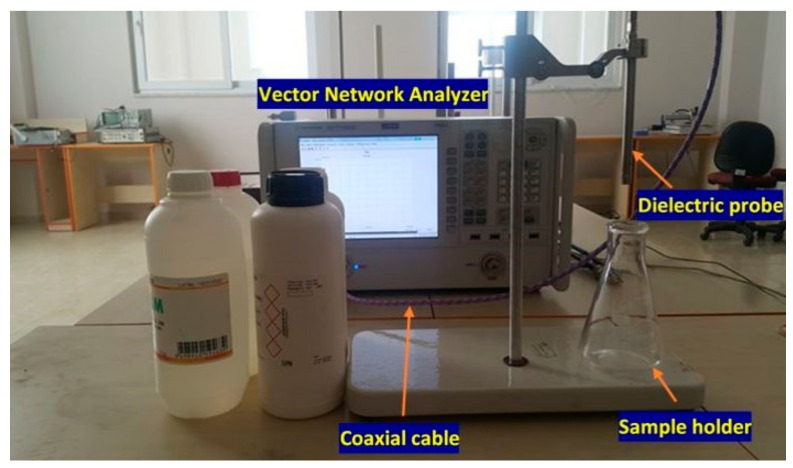
Experimental setup to determine dielectric measurement of the chemical samples by 85070E dielectric probe kit.

**Figure 8 sensors-20-00943-f008:**
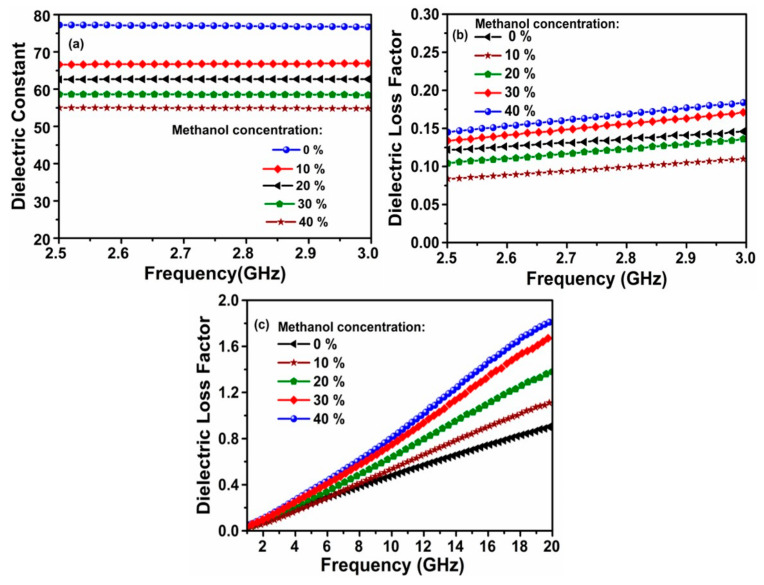
Measured results for methanol–water mixture: (**a**) the dielectric constant; (**b**) dielectric loss at 2.5 -3 GHz, and (**c**) dielectric loss factor at 1–20 GHz frequency band.

**Figure 9 sensors-20-00943-f009:**
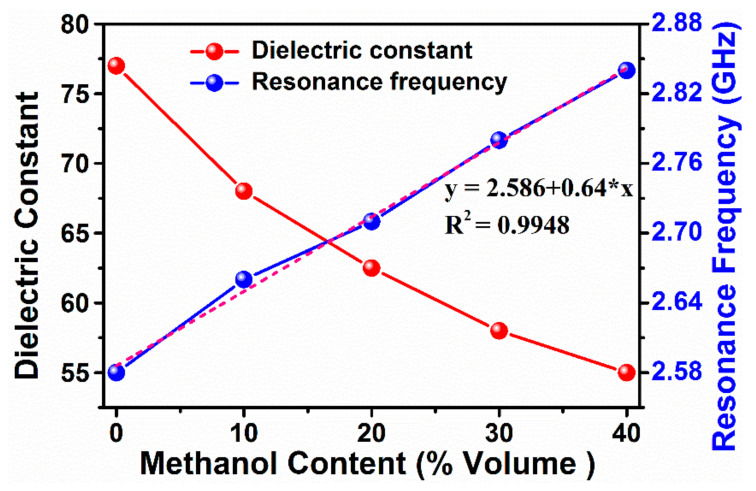
Dependence of resonance frequency on methanol volume concentration.

**Figure 10 sensors-20-00943-f010:**
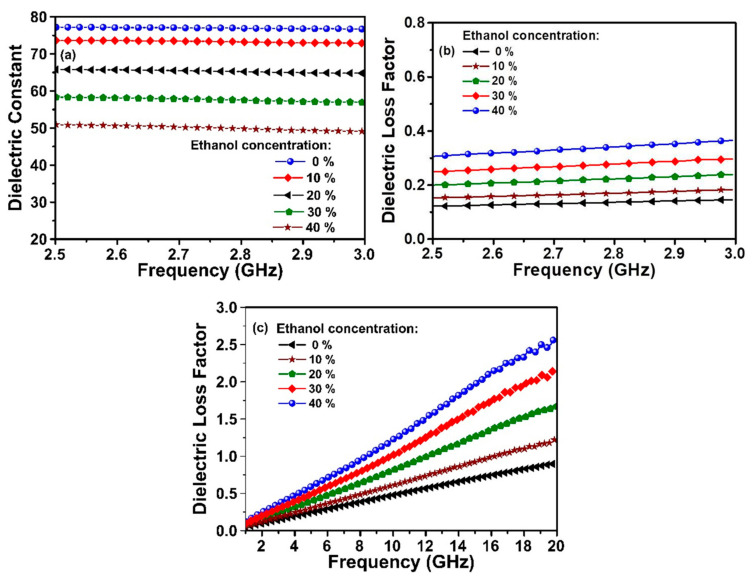
Measured results for ethanol–water mixture (**a**) the dielectric constant (**b**) dielectric loss factor at 2.5–3 GHz (c) dielectric loss factor at 1–20 GHz frequency band.

**Figure 11 sensors-20-00943-f011:**
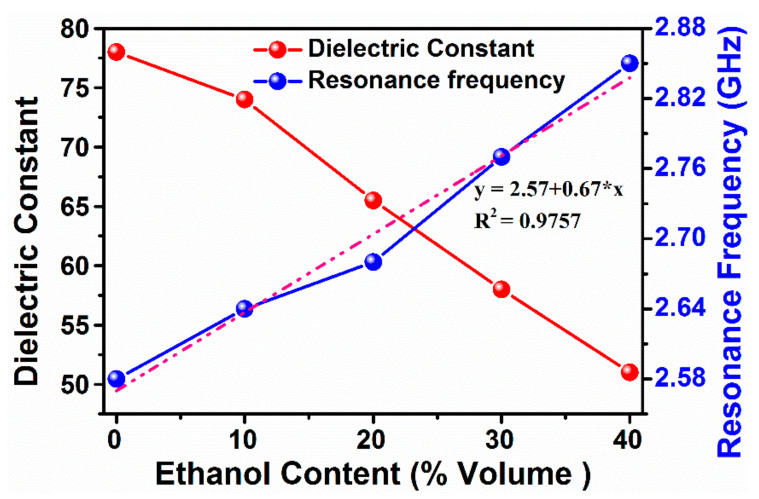
Simulated results variation of resonance frequency with ethanol content and dielectric constant linearity dependent.

**Figure 12 sensors-20-00943-f012:**
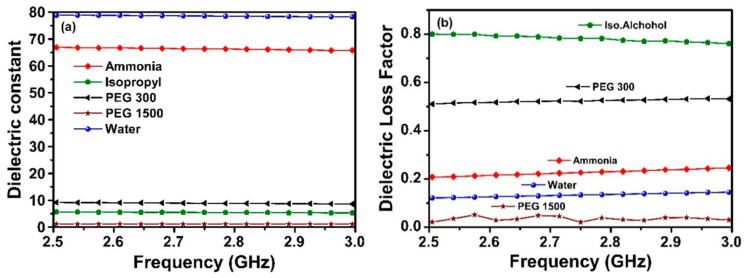
Measured results for different samples: (**a**) dielectric constant; (**b**) dielectric loss factor.

**Figure 13 sensors-20-00943-f013:**
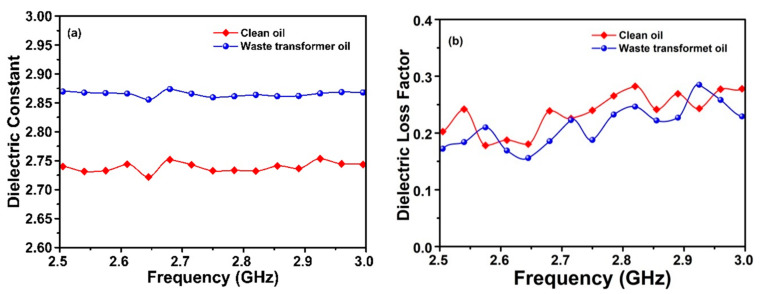
Measured results for the clean and waste transformer oils (**a**) dielectric constant; (**b**) dielectric loss factor.

**Figure 14 sensors-20-00943-f014:**
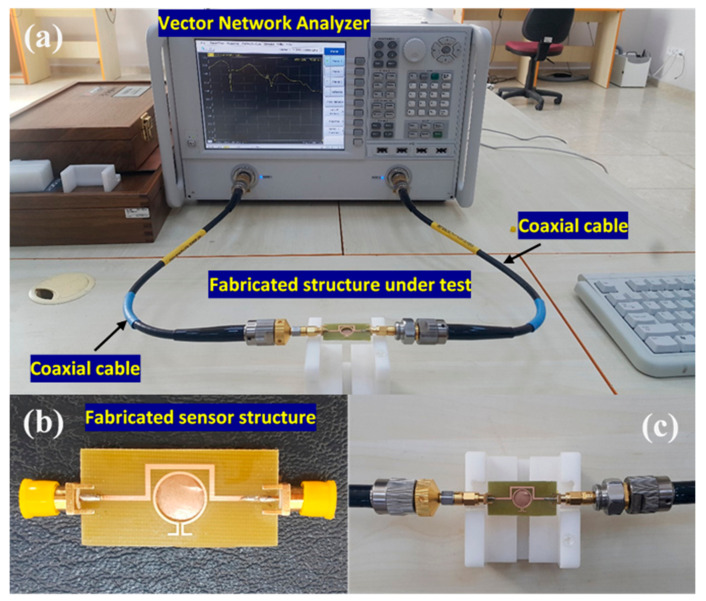
(**a**) Experimental measurement setup, (**b**) fabricated proposed structure; (**c**) fabricated structure connected to the vector network analyzer.

**Figure 15 sensors-20-00943-f015:**
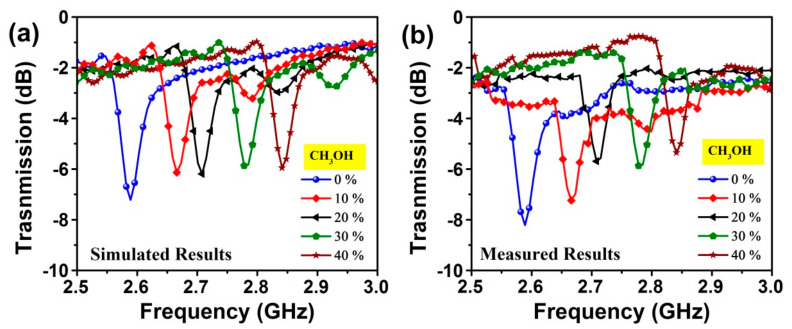
Effect of methanol (CH_3_OH) concentration on the transmission coefficient over 2.5- 3 GHz: (**a**) simulated; (**b**) measured results.

**Figure 16 sensors-20-00943-f016:**
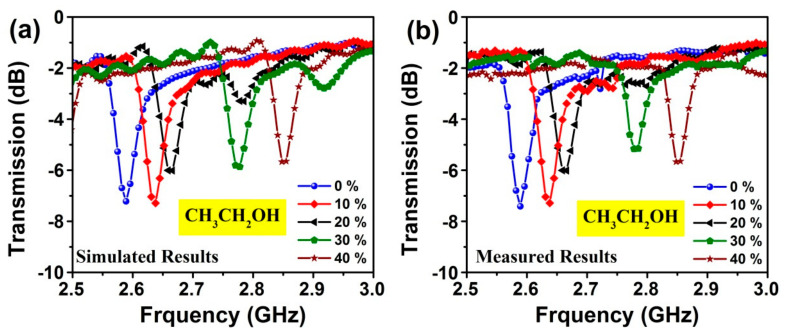
Dependence of transmission coefficient on the ethanol (CH_3_CH_2_OH) concentration over 2.5–3 GHz: (**a**) simulated; (**b**) measured results.

**Figure 17 sensors-20-00943-f017:**
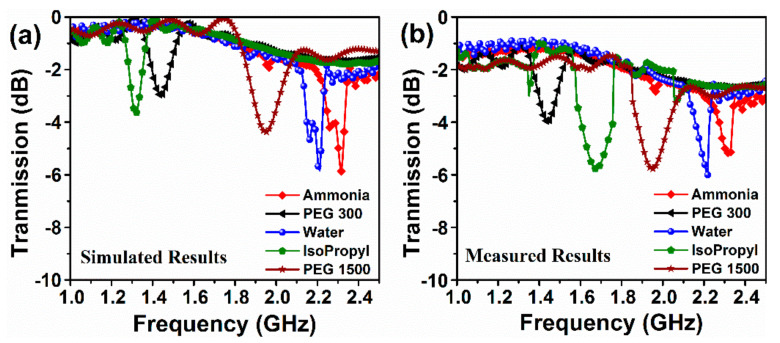
Transmission coefficient of the different pure chemical liquids over 2.5–3 GHz: (**a**) simulated; (**b**) measured results.

**Figure 18 sensors-20-00943-f018:**
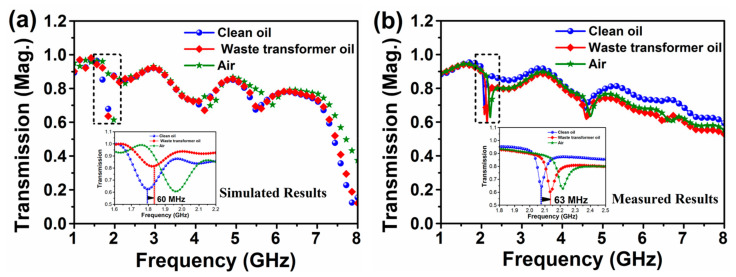
Transmission coefficient for the clean and waste transformer oils at frequency 1–8 GHz (**a**) simulated result; (**b**) fabricated result.

**Table 1 sensors-20-00943-t001:** Comparison of sensitivity between our proposed structure and other in literature.

Reference	Dielectric Constant	Frequency Range (GHz)	Resonant Frequency Shift (MHz)
[19]	66 for 10% methanol and 77 for the 10% ethanol	4–5	35 for methanol and 30 for ethanol
[20]	39 for 40% ethanol	1.3–2.3	40
[21]	58 for methanol 40% and 57 for 40% ethanol	1.7–2.1	30 for methanol and 15 for ethanol
[25]	55 for 30% ethanol	1–3	90
[26]	57 for 40% methanol and 53 for 40% ethanol	0.8–2.2	20 for methanol and 30 for ethanol
[27]	76.84 for water, 6.62 for ethanol and 20.54 for methanol	2, 5, and 7	8 for water, 2 for methanol, 3 for ethanol
[28]	59 for 40% ethanol	0.8–0.95	10
This work	77.5 for water, 57 for 40% methanol and 56 for 40% Ethanol	2.5–3.5	100 for water, 90 for methanol, and 80 for ethanol

**Table 2 sensors-20-00943-t002:** Comparison of sensitivity for the rest chemical samples between our proposed structure and other in literature.

Reference	Dielectric Constant	Frequency Range (GHz)	Resonant Frequency Shift (MHz)
[19]	17 for ammonia and 9.5 for PEG 300	4–6	5 for ammonia and 5 for PEG 300
This Work	67 for ammonia and 9.5 for PEG 300	2.5–3	20 for ammonia and 10 for PEG 300
[34]	2.7 and 2.9 for clean and waste oil	2–6	40
This Work	2.74 and 2.87 for clean and waste oil	1–8	63

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
