# Peer review of "Novel Metamaterials-Based Hypersensitized Liquid Sensor Integrating Omega-Shaped Resonator with Microstrip Transmission Line"

_sensors, 2020, doi:10.3390/s20030943_

Round 1
Reviewer 1 Report
Add the model establishment and index optimization to analysis of the key dimensions of the sensor structure in this work. Increase the effect comparison with closely related sensors in this field, or index analysis that can clearly indicate the advantages of the sensor structure in this work. Describe the working principle of the sensor that intergrates resonator and microstrip transmission line in more detail. Organize the article structure more clearly, such as,it is difficult to understand that put the simulation part of sensor structure in chapter 4. Carefully check the language of the article. There are some problems, such as inconsistant experssions, such as, “metamaterial-based sensor”and“MTM integrated transmission line-based sensor” are both considered as the sensor proposed in the paper, but with different names. Revise the chart of the article. There are problems of inconsistent colors of Fig 13 and unclear display of Fig 14.
Reviewer 2 Report
Please find the comments in the attached file.

Round 2
Reviewer 1 Report
1.For better legibility, check consistency of charts clarity, e.g. Compared Figure 8,10,12,13 with Figure 5.
2.Further checking for typos, e.g. “Qu factor” in Page 3.
Author Response
Reviewer 1:
For better legibility, check consistency of charts clarity, e.g. Compared Figure 8,10,12,13 with Figure 5.Response: Thank you for your suggestion. We have carefully redrawn all figures for better legibility in our revised manuscript.
Further checking for typos, e.g. “Qu factor” in Page 3.Response: Thank you for your attention. It should be “Q factor”.
Reviewer 2 Report
Please see the attached file.

Author Response
Reviewer 2
The authors have considered all the previous comments in the revised manuscript and have provided satisfactory answers, and the amount of work devoted is noticeable. As a consequence, this reviewer feels that the quality of the paper has been considerably increased, especially regarding the experimental results and the comparison with other similar works. The results are relevant and the discussion is appropriate, and thus the paper is now suitable for acceptance, provided that the authors consider the following
Minor aspects:
There is a strong need for a comprehensive English checking, especially in the newly added parts.Response: Thank you for your comments. We have carefully revised expressions throughout this manuscript.
Check the frequency range in line 84 (it is unlikely 207 GHz).Response: Thank you for your attention. The operating frequencies are 2 GHz, 5 GHz, and 7 GHz.
Figs. 15, 16 and 17: use always the same legend format.Response: Thank you for your attention. Actually, Figure 15 and 16 illustrate transmission coefficient of sensor filled with methanol (CH3OH) and ethanol (CH3CH2OH) at different concentration, while Figure 17 shows transmission coefficient when using typical pure chemical liquids (i.e., PEG 300, Isopropyl, PEG1500, Ammonia and water). Legends for these pictures are revised.
Fig. 17: say which is Fig. (a) and (b) in the caption.Response: Thank you for your attention. We have added (a) Simulated and (b) Measured results in the in the revised manuscript.
The second author’s names are wrong in references [25, 26, 27, 28, 29], they should be fixed. Please look the references provided in the comments in the 1st round.Response: Thank you for your attention! All author’s names are carefully updated in the revised manuscript.
Tables 1 and 2 really enhance the quality of the manuscript. It would be even more complete if comparison with reference [27] is also included in Table 1. Please note that it is “comparison”, not “compression” in the captions.Response: Thank you for your positive comment, the comparison sensitivity of this work with previous work is added in Table 1 in the revised manuscript.
Check the term “g-shaped resonator” in Fig. 5 caption.Response: Thank you for your attention, we are sorry for our mistake. It should be “omega-shape”.
In the definition of the permittivity, in Eq. (4), by convention it is more common to have the imaginary part negative: ɛ’–ɛ’’Response: Thank you for your comment, we changed the imaginary part as you mentioned.